# Effects of Different Processing Methods Based on Different Drying Conditions on the Active Ingredients of *Salvia miltiorrhiza* Bunge

**DOI:** 10.3390/molecules27154860

**Published:** 2022-07-29

**Authors:** Liuwei Zhang, Xuemei Zhang, Naheeda Begum, Pengguo Xia, Jingling Liu, Zongsuo Liang

**Affiliations:** 1College of Chemistry and Pharmacy, Northwest A&F University, Yangling 712100, China; zlw15877395210cpyy@126.com (L.Z.); 18404967815@163.com (X.Z.); 2Shandong (Linyi) Institute of Modern Agriculture, Zhejiang University, Linyi 276000, China; 3National Center for Soybean Improvement, Key Laboratory of Biology and Genetics and Breeding for Soybean, Ministry of Agriculture, State Key Laboratory of Crop Genetics and Germplasm Enhancement, Nanjing Agricultural University, Nanjing 210095, China; T2020106@njau.edu.cn; 4College of Life Science and Medicine, Zhejiang Sci-Tech University, Hangzhou 310018, China; xpg_xpg@126.com; 5College of Life Science, Northwest A&F University, Yangling 712100, China; jinglingliu-sm@Nwsuaf.edu.cn

**Keywords:** processing method, drying condition, active ingredients, *Salvia miltiorrhiza*

## Abstract

Compared to the traditional processing method, fresh processing can significantly enhance the preservation of biologically active ingredients and reduce processing time. This study evaluated the influences of fresh and traditional processing based on different drying conditions (sun drying, oven drying and shade drying) on the active ingredients in the roots and rhizomes of *S. miltiorrhiza*. High-performance liquid chromatography (HPLC) was utilized to determine the contents of six active ingredients in the roots and rhizomes of *S. miltiorrhiza*. The data were analyzed by fingerprint similarity evaluation, hierarchical cluster analysis (HCA) and principal component analysis (PCA). The results suggest that compared to the traditional processing method, the fresh processing method may significantly increase the preservation of biologically active ingredients. Furthermore, the findings demonstrated that among the three drying methods under fresh processing conditions, the shade-drying (21.02–26.38%) method is most beneficial for retaining the active ingredients in the roots and rhizomes of *S. miltiorrhiza*. Moreover, the fingerprint analysis identified 17 common peaks, and the similarity of fingerprints among samples processed by different methods ranged from 0.989 to 1.000. Collectively, these results suggest novel processing methods that may improve the yield of active ingredients for *S. miltiorrhiza* and may be implemented for industrial production.

## 1. Introduction

*Salvia miltiorrhiza* Bunge, from the Lamiaceae family of plants, is a well-known traditional Chinese medicine (TCM) [1] (Figure 1). The dried roots and rhizomes of *S. miltiorrhiza* promote blood circulation, relieve dysmenorrhea and reduce swelling [2]. More than 70 compounds in *S. miltiorrhiza* have been isolated and identified [3,4]. There are two groups of main bioactive ingredients: water-soluble phenolic acids, such as rosmarinic acid (Figure 2A) and salvianolic acid B (Figure 2B), and lipid-soluble terpenoid ingredients, such as dihydrotanshinone I (Figure 2C), cryptotanshinone (Figure 2D), tanshinone I (Figure 2E) and tanshinone IIA (Figure 2F) [5,6,7]. Recent studies have demonstrated that tanshinones have various biological functions, including antitumor, antibacterial, antioxidant and anti-inflammatory activities [8,9], and tanshinones are widely employed to treat multiple cardiovascular and cerebrovascular diseases and breast cancer [10,11,12,13]. Furthermore, hydrophilic phenolic acid derivatives have vital roles in the treatment of cardiovascular diseases, inflammation [14,15,16], angina pectoris and dysmenorrhea [17,18], due to their antithrombic, anticoagulation, cytoprotective, antiviral, anti-inflammatory and antioxidant effects [19,20,21]. In addition, it was recently reported that salvianolic acid B can inhibit the viropexis of 2019-nCoV spike pseudovirus [22]. Due to the excellent curative effects of *S. miltiorrhiza*, it is widely applied as a raw material in Chinese patent medicine, such as compound *S. miltiorrhiza* tablets and dripping pills, and in food additives and cosmetics in Asian countries [23]. Consequently, the demand for *S. miltiorrhiza* has steadily increased in recent years. China consumes an estimated 20 million kilograms of *S. miltiorrhiza* annually [24,25,26]. Due to the low contents of active ingredients, long growth cycle and severe quality degradation of *S. miltiorrhiza* [27], enhancement of the contents of active ingredients in *S. miltiorrhiza* has recently become an active area of research. 

Many factors affect the quality of TCMs, such as growth years, harvest time, pest control, germplasm quality and postharvest processing [28,29,30]. Among them, postharvest processing is one of the most important factors. Postharvest processing may lead to changes in active ingredients that directly affect clinical efficacy and stability [31]. It has been recently reported that in the traditional postharvest processing of *S. miltiorrhiza*, its roots and rhizomes are dried after harvesting in the production area. After entering the market as a TCM, the roots and rhizomes are washed, softened, cut and then dried again. It thus undergoes two drying steps (a first drying step after harvesting and a second after sectioning) and a softening process. Previous findings demonstrated that the whole process is time-consuming and laborious, and the active ingredients are easily lost in the softening process [32]. Recently, it was reported that salvianolic acid B and rosmarinic acid are very soluble in water and that a longer exposure time of *S. miltiorrhiza* to water during the softening process leads to a more severe loss of ingredients [33]. Therefore, it is necessary to optimize the traditional processing technology of *S. miltiorrhiza* to reduce the loss of water-soluble phenolic acids during postharvest processing. A general approach has been to reduce the time of contact with water as much as possible. One proposed strategy is to cut fresh *S. miltiorrhiza* into thick pieces postharvest in the planting area [34]. In recent years, fresh processing technology for some Chinese medicines, such as *Panax notoginseng* (Burk.) F.H. Chen [35], *Scutellaria baicalensis* Georgi [36] and *Angelica sinensis* [37], has been explored. 

Drying is one of the most important processing methods for TCMs. Drying can decrease the volume and weight of TCMs and reduce the costs of storage, transportation and packaging [38]. More importantly, it may maintain active ingredients and extend the preservation time of TCMs [39,40,41]. However, the contents and species of active ingredients can be modified during the postharvest drying period due to anti-dehydration stress [42]. There are numerous drying methods for TCMs, including microwave drying, freeze drying, steam drying, shade drying, sun drying, vacuum drying, etc. [43,44,45,46,47]. Different drying methods have different effects on the active ingredients of TCMs [48]. Sun drying, oven drying and shade drying have been reported to be the most common drying methods in the postharvest processing of TCMs [49,50]. However, comprehensive research on the processing of *S. miltiorrhiza* by different drying methods is lacking, and studies on the fresh processing and traditional processing of *S. miltiorrhiza* based on different drying conditions have not been reported. Thus, the differences in active ingredient contents and overall quality among *S. miltiorrhiza* materials processed with different methods remain largely unknown.

The aim of our study was to evaluate the differences between fresh processing and traditional processing, and demonstrate the feasibility of fresh processing by using HPLC to determine the contents of the active ingredients of *S. miltiorrhiza* processed via different methods. Meanwhile, the differences in the contents of active ingredients among different drying conditions under fresh processing were studied. In addition, to further study the impacts of different processing methods on *S. miltiorrhiza*, we carried out fingerprinting evaluation, HCA and PCA. 

## 2. Results

### 2.1. Effects of Fresh and Traditional Processing Methods on the Contents of the Main Active Ingredients in the Roots and Rhizomes of S. miltiorrhiza

To investigate the effects of fresh and traditional processing on the active ingredients of *S. miltiorrhiza* roots and rhizomes, HPLC was employed to determine the contents of the active ingredients under fresh and conventional processing for each drying method (Table 1). In the samples processed by the shade-drying method, the content of total tanshinones in S2 was lower than that in S14, and the contents of salvianolic acid B, rosmarinic acid and total tanshinones in S1, S3, S4 and S5 were significantly higher than those in S14 (*p <* 0.05, Figure 3A), except for total tanshinone content in S1. The contents of salvianolic acid B, rosmarinic acid and total tanshinones in samples processed by the sun-drying method were significantly higher in S1, S6, S7, S8 and S9 than those in S14 (*p <* 0.05, Figure 3B). For the samples treated by the oven-drying method, the contents of salvianolic acid B and rosmarinic acid in S1, S10, S11, S12 and S13 were significantly higher than that in S16 (*p <* 0.05, Figure 3C). For the total tanshinones, there was no significant differences between S1 or S10 and S16, whereas the contents of S11, S12 and S13 were significantly higher than those of S16 (*p <* 0.05, Figure 3C). These results demonstrated that the fresh processing method could retain much of the contents of the active ingredients and was superior to traditional processing. 

### 2.2. Differences in the Contents of the Main Active Ingredients among Different Drying Conditions under Fresh Processing (S1–S13)

To further reveal the effect of drying conditions on the active ingredients under fresh processing, we analyzed the differences in active ingredients among the conditions of shade drying, sun drying and oven drying (60 °C) with different moisture contents. When the moisture content was high (S1,S2, S1–S6, S1–S10), the content of salvianolic acid B decreased in the order shade drying, sun drying and oven drying. In shade-dried (S2–S5), sun-dried (S6–S9) and oven-dried (S10–S13) samples, with the extension of drying time, moisture content gradually decreased, and the content of salvianolic acid B increased (Figure 4A). For rosmarinic acid, when the moisture content was high (S1,S2, S1–S6, S1–S11), the content of rosmarinic acid decreased in the order shade drying, sun drying and oven drying. In shade-dried (S2–S5), sun-dried (S6–S9) and oven-dried (S11–S13) samples, with the extension of drying time, moisture content gradually decreased, and the content of rosmarinic acid increased (Figure 4B). The contents of the shade-dried (S4,S5) and sun-dried (S8,S9) samples were greater than those of the other processed samples. However, for the samples processed by the oven-drying method, which yielded the highest contents of rosmarinic acid and salvianolic acid B (S13), the differences from the undried sample (S1) were not significant. These results indicated that for the fresh processing method, shade drying (S4,S5) and sun drying (S8,S9) were beneficial for preserving the contents of rosmarinic acid and salvianolic acid B, while oven drying was not.

For total tanshinones, with the extension of drying time, the moisture contents gradually decreased. The content of total tanshinones under the shade drying and oven drying methods showed irregular trends. However, compared to that in S1, the contents of total tanshinones in the shade-dried (S3–S5) and oven-dried (S11–S13) samples were high. During the sun-drying process, the contents of total tanshinones continuously decreased (S1–S8); although the content of total tanshinones (S8,S9) increased, it remains less than that in S1 (Figure 4C). The contents of the shade-dried (S3–S5) and oven-dried (S11–S13) samples were greater than those of the other processed samples. However, the highest total tanshinone content was observed in the sun-dried samples (S6), although it was lower than that of the undried sample (S1). The results suggested that for the fresh processing method, shade drying (S3–S5) and oven drying (S11–S13) were beneficial for preserving the contents of total tanshinones, while sun drying was not.

Therefore, in the fresh processing method, as moisture content decreased over the drying process, the contents of salvianolic acid B, rosmarinic acid and total tanshinones changed (increased or decreased). From these results, we inferred that the shade-drying method (S4–S5) might be the best drying method for preserving the main active ingredients. 

### 2.3. HPLC Fingerprinting and SA of S. miltiorrhiza Treated by Different Processing Methods 

To further comprehensively explore the effects of different processing methods on *S. miltiorrhiza*, HPLC fingerprints of the sixteen groups of *S. miltiorrhiza* processed via different drying and cutting methods were established. The chromatographic data were analyzed using the software Similarity Evaluation System for Chromatographic Fingerprints of Traditional Chinese Medicine (Version 2012). There were a total of 17 common peaks (accounting for more than 90% of the total PA) in the HPLC fingerprint, which appeared in the 16 samples (Appendix A, see Appendix A); the six reference chromatograms are shown in Figure 5A. By comparing the t_R_ and HPLC-UV spectrum of each peak with those of the mixed standard compound, the six ingredients were marked as peak 7 (rosmarinic acid), peak 9 (salvianolic acid B), peak 12 (dihydrotanshinone I), peak 14 (cryptotanshinone), peak 15 (tanshinone I) and peak 17 (tanshinone IIA). Peak 17 (tanshinone IIA), with a large PA and stable peak time, was assigned as the reference peak to calculate the relative retention area and relative t_R_ of the other sixteen common peaks. As shown in Appendix A, the relative t_R_ R.S.D. of each common peak ranged from 0.036–1.640%, and the relative PA R.S.D. ranged from 7.772–64.680%. There were significant differences in the content of each ingredient of *S. miltiorrhiza* among different processing methods. The similarity values are shown in Appendix A. The similarity values of the 16 batches of samples with the reference fingerprint ranged from 0.989–1.000, which showed that all samples had high similarity. Based on these findings, it was concluded that the different treatment methods did not significantly affect the category of bioactive ingredients of *S. miltiorrhiza*.

### 2.4. HCA of S. miltiorrhiza

HCA is a multivariate analysis method that provides visual information about raw data [51]. The relative PAs of the 17 common peaks of 16 batches of *S. miltiorrhiza* samples were imported into IBM SPSS 26 software to perform cluster analysis with the between-group connection method and the Euclidean square distance as the classification basis. As shown in Figure 6, the 16 batches of samples could be divided into three groups. Samples S1, S2, S3, S6, S7, S10, S11, S12, S13 and S14 were assembled into the first group; samples S4, S5, S8 and S9 were clustered into the second group; and S15, S16 were clustered into the third group. The groups were classified according to the contents of the active ingredients (Table 1). The sample sources comprised one-year-old *S. miltiorrhiza* plants from Tongchuan, but they belonged to different types, which may have been related to the different processing methods of TCM. The findings showed that the type of processing method can affect the quality of the TCM.

### 2.5. PCA of S. miltiorrhiza 

PCA is a commonly used method of multivariate analysis and is usually employed in HPLC fingerprinting research [52,53]. In this study, the PA of 17 unique peaks in 16 groups of samples, as described above, was submitted to SPSS 26 for PCA. The best principal components were defined as those with eigenvalues λ > 1, and four principal components were extracted. The eigenvalues of the first four principal components were 6.843, 4.447, 1.902 and 1.452, and the cumulative contribution rate was 86.145% (Appendix A), which satisfied the requirement of cumulative percentage of variance (CPV) > 70–85% for PCA. Therefore, the four principal components contained most of the information in the variables. PC1 had positive factor loadings for 10 peaks: A4, A7, A9, A10, A11, A12, A13 (cryptotanshinone), A14 (tanshinone I), A16 (tanshinone IIA) and A17 (salvianolic acid B). PC2 had positive factor loadings for 8 peaks: A1, A2, A6, A12, A13 (cryptotanshinone), A14 (tanshinone I), A15 and A16 (tanshinone IIA) (Appendix A). Accordingly, the score plot of the first and second principal components, PC1 and PC2, showed apparent differences among all samples (S1–S16). As shown in the score plot of the PCA results (Figure 7), differences between S10–S13 and S6–S9 are evident on both the positive and negative sides. S16 and S14 are more closely located. S2–S5 are on the right side except for S2. The PCA clearly demonstrates substantial differences between the oven-dried and sun-dried samples, and differences among drying methods and even among moisture contents within a drying method.

## 3. Discussion 

### 3.1. Fresh Processing May Be Widely Adopted for S. miltiorrhiza Processing in the Future

In the present study, we compared fresh and traditional methods of processing *S. miltiorrhiza.* The investigation revealed that the fresh processing method is favorable for the preservation of salvianolic acid B and rosmarinic acid. Fresh processing can avoid the washing and softening steps of traditional processing, thereby reducing the damage to the water-soluble ingredients salvianolic acid B and rosmarinic acid. For preserving tanshinones, fresh processing was generally better than traditional processing. The fingerprint SA showed that the fingerprints of the traditional and fresh processing methods are very similar, which indicates that the types of 17 active ingredients are not substantially changed under fresh processing compared with traditional processing. In general, the new technology of fresh processing offers cost and energy savings, and can maximize the retention of active ingredients, and it has fundamental significance for the processing of TCMs. Fresh processing technology for *S. miltiorrhiza* may become a mainstream processing technology in the future. 

### 3.2. The Contents of Active Ingredients Changed during Processing

Changes in the contents of active ingredients during TCM processing have become a very important issue. Previous studies have suggested that the active ingredients of medicinal plants form and accumulate before harvest and that their contents decrease following harvest [54]. In the present research, we investigated whether the contents of total tanshinones, salvianolic acid B and rosmarinic acid in newly harvested plant roots and rhizomes increased after harvest. Consistent with our findings, other studies have revealed significant increases in the contents of total tanshinones and salvianolic acid B in the roots of *S. miltiorrhiza* in the early drying process after harvest [55,56]. One explanation for this phenomenon is that newly harvested plant materials (especially roots) still have physiological activity and that the antidehydration response of the plant is induced in the early stage of drying and dehydration after harvest [57,58]. By inducing endogenous antioxidant enzymes to produce active ingredients, the antioxidant activity of plants can be improved [59,60,61]. As both salvianolic acid B and total tanshinones in *S. miltiorrhiza* are effective oxygen free radical (OFR) scavengers, they play important roles in the resistance of plants to dehydration stress [62]. Therefore, the total tanshinone, salvianolic acid B and rosmarinic acid contents in *S. miltiorrhiza* might result from the remobilization and transformation of preharvest stored carbohydrates in roots and rhizomes by a series of primary and secondary metabolic reactions under dehydration stress.

### 3.3. Effects of Different Drying Conditions on the Contents of Salvianolic Acid B and Rosmarinic Acid during Fresh Processing

For the fresh processing method, shade drying (S4–S5) and sun drying (S8–S9) were beneficial for preserving the contents of rosmarinic acid and salvianolic acid B, while the other treatments were not. This result may be related to the enzymes in *S. miltiorrhiza*. Polyphenol oxidase (PPO) is widely distributed in plant cells and is a key enzyme causing browning [63]. There are three conditions for enzymatic browning: substrate, oxygen and enzyme. Under normal circumstances, PPO cannot directly contact phenolic substances due to the regional distribution of the membrane system. PPO primarily exists in the cytoplasm, while phenols exist in vacuoles. Therefore, enzymatic browning will not occur even in the presence of oxygen [64]. However, when *S. miltiorrhiza* is cut under fresh conditions, a large number of phenolic acids in vacuoles come into contact with PPO due to cell structure damage [65]. In the presence of oxygen, PPO can oxidize o-diquinone in phenolic acids by molecular oxygen [66], resulting in decreased contents of salvianolic acid B and rosmarinic acid. Therefore, it is speculated that in the present study, regardless of the drying method, PPO enzyme activity was very high when the water content was high (greater than 34%). When the water content was below 26%, PPO activity may have been greatly reduced, limiting the oxidation of salvianolic acid B and rosmarinic acid. Moreover, compared with the shade drying and sun drying methods, oven drying at 60 °C was not favorable to the preservation of rosmarinic acid and salvianolic acid B; in this process, *S. miltiorrhiza* was cut into thick slices regardless of water content, and the contents of active ingredients after drying were very low. Rosmarinic acid and salvianolic acid B have catechol and Ar-CH=CH-COOR structures, which are easily oxidized at high temperatures [67]. During the first oven-drying process, the temperature was not very high, but with the extension of drying time, the roots and rhizomes were also not cut into thick slices, resulting in continuous heating inside the roots and rhizomes. The internal heat was not dispersed, creating strong conditions for a series of complex enzymatic, oxidation, polymerization and other biochemical reactions, so the contents of the oven-dried samples were consistently very low. Therefore, the use of the shade-drying method (moisture content: 21.02–26.38%) or the sun-drying method (moisture content: 13.88–22.91%) may reduce the damage to rosmarinic acid and salvianolic acid B. 

### 3.4. Effects of Different Drying Conditions on Total Tanshinone Contents during Fresh Processing 

The contents of total tanshinones under the sun-drying method were significantly lower than those under the other two methods. Total tanshinones and other fat-soluble active ingredients are distributed around the periderm of the roots and rhizomes of *S. miltiorrhiza* [68]. Therefore, total tanshinone ingredients are sensitive to light and unstable and are thus easily decomposed under light conditions. In the sun-drying process, the materials were exposed to the sun for a long time, and due to the existence of autogenous enzymes, some functional ingredients easily decomposed, causing a severe loss of total tanshinones. However, the use of the shade-drying method (moisture content: 21.02–34.57%) or the oven-drying method (moisture content: 18.01–49.79%) may reduce the damage to total tanshinones.

## 4. Materials and Methods

### 4.1. Chemicals and Plant Materials 

HPLC-grade acetonitrile was purchased from Merck Co., Inc. (Darmstadt, Germany). HPLC-grade phosphoric acid and analytical-grade methanol were purchased from Tianjin Komiou Chemical Reagent Co., Ltd. (Tianjin, China). Six reference compounds (rosmarinic acid, salvianolic acid B, dihydrotanshinone I, cryptotanshinone, tanshinone I and tanshinone IIA, HPLC ≥ 98%) were purchased from Shanghai Yuan ye Bio-Technology Co., Ltd. (Shanghai, China). Ultrapure water was generated with a You pu Ultrapure Water System (Chengdu, China).

The fresh roots and rhizomes of one-year-old *S. miltiorrhiza* plants were harvested in 2020 in Tongchuan (Tongchuan, China) in April, the best time of year for the harvest of the medicinal parts of *S. miltiorrhiza*, and were identified by Professor Zongsuo Liang (Zhejiang Sci-tech University). After removing soil and impurities, the samples were divided into 16 batches according to their weight, size and thickness.

### 4.2. Sample Processing

#### 4.2.1. Fresh Processing

The roots and rhizomes of the freshly harvested *S. miltiorrhiza* plants were divided into 13 groups (Codes S1–S13). As shown in Table 2, different drying conditions were applied. *S. miltiorrhiza* roots and rhizomes were dried for different durations to achieve different moisture contents (Table 2). Then, the samples (S1–S13) were cut into 2–4 mm thick slices, and the slices were dried in an oven at 60 °C until they were completely dry (≤13.0%). All treatments were replicated three times (Figure 8).

#### 4.2.2. Traditional Processing

The roots and rhizomes of the freshly harvested *S. miltiorrhiza* plants were divided into 3 groups (codes S14–S16). As shown in Table 1, different drying conditions were applied. *S. miltiorrhiza* roots and rhizomes were dried until they were completely dry (≤13.0%, Table 2). Then, after being washed and softened, the samples (S14–S16) were cut into 2–4 mm thick slices, and the slices were dried in an oven at 60 °C until they were completely dry (≤13.0%). All treatments were replicated three times (Figure 8).

#### 4.2.3. Shade Drying

This method was carried out in a dark room. The S2–S5 and S14 samples were spread out in a dark, ventilated room without direct sunlight. The average temperature of the room was 13.5 °C, and the relative humidity was 55% ± 5%.

#### 4.2.4. Sun Drying

This method was performed in an open environment. S6–S9 and S15 samples were placed on a hardboard with exposure to direct sunlight. The average temperature was 16.8 °C (mid-late March to mid-April).

#### 4.2.5. Oven Drying

An electric constant-temperature blast-drying oven (DHG-9240A, Shanghai Jing Hong Laboratory Instrument Co., Ltd., Shanghai, China) was used to dry samples (S10–S13, S16) at 60 °C.

### 4.3. Preparation of Samples and Reference Standards

The dried samples were sieved through a Standard Chinese Pharmacopeia sieve-3 (355 ± 13 μm). A total of 0.5 g of each sample was accurately weighed and then placed in a 50 mL centrifuge tube. After immersion in 70% methanol for approximately 12 h, the sample was subjected to ultrasonic extraction with 20 mL of 70% methanol for 60 min at room temperature and centrifuged for 25 min at 7830 rpm. The supernatant was filtered through a 0.22 μm membrane. The samples were stored in a refrigerator at 4 °C before HPLC-PDA analysis.

An appropriate amount of each of six reference standards of rosmarinic acid, salvianolic acid B, dihydrotanshinone I, cryptotanshinone, tanshinone I and tanshinone IIA was dissolved in methanol (99.8%) to obtain the desired final contents (715.00, 3360.00, 94.500, 102.50, 135.50 and 215.00 µg·mL^−1^ of the mixed reference standard solution), and the mixed solution was sealed and stored in a refrigerator at 4 °C.

### 4.4. Instrumentation and HPLC Conditions

Fingerprinting analysis of *S. miltiorrhiza* was performed on a Waters 1525 HPLC system (Waters Corp., Milford, MA, USA) consisting of a binary pump, a 2996 photodiode array detector, a column temperature controller and a manual injector. Data acquisition and system control were supported by Empower 2 software. Chromatographic separation was performed on a Waters Symmetry C18 column (4.6 mm × 250 mm, 5 µm). A gradient elution of A (acetonitrile) and B (0.02% phosphoric acid solution) was utilized as follows: 0–10 min: 5–20% A; 10–15 min: 20–25% A; 15–20 min: 25% A; 20–25 min: 25–20% A; 25–28 min: 20–30% A; 28–40 min: 30% A; 40–45 min: 30–45% A; 45–50 min: 45–50% A; 50–58 min: 50–58% A; 58–67 min: 58–55% A; 67–70 min: 55–60% A; 70–80 min: 60–65% A; 80–87 min: 65% A; 87–91 min: 65–100% A. The flow rate was set to 1 mL·min^−1^, and the column temperature was set to 30 °C. The chromatographic peak area (PA) was monitored via the HPLC-PDA detector at wavelengths of 288 nm (rosmarinic acid, salvianolic acid B) and 270 nm (dihydrotanshinone I, cryptotanshinone, tanshinone I and tanshinone IIA). The peak information at 270 nm was applied for HPLC fingerprinting analysis.

### 4.5. Determination of Water Content in Sample

The moisture content was determined according to the drying method in the fourth part of the Chinese Pharmacopoeia of the 2020 version [69].

### 4.6. Main Peak Identification and Sample Content Determination

Under the HPLC conditions described in Section 2.4, the reference standard solution was injected into the HPLC system, and the t_R_ and PA of the reference standard were recorded. The active ingredients of all samples were identified by comparing the t_R_ values with those of the reference standards. The contents of the active ingredients were calculated according to the reference standard contents and PAs.

### 4.7. Similarity Evaluation and Statistical Analysis

HPLC fingerprint similarity analysis (SA) was evaluated with Similarity Evaluation System for Chromatographic Fingerprint (Version 2012) software, which was developed and recommended by the Chinese Pharmacopoeia Committee. HCA and PCA of all samples were performed using the Statistical Program for Social Sciences (SPSS) 26 software (SPSS Inc., Cary, NC, USA).

## 5. Conclusions

In this study, fresh processing and traditional processing methods based on different drying conditions were utilized to explore their effects on the active ingredients in the roots and rhizomes of *S. miltiorrhiza.* We used HPLC fingerprinting, SA, HCA and PCA to examine the differences in salvianolic acid B, rosmarinic acid and total tanshinone contents among *S. miltiorrhiza* roots and rhizomes processed by different methods. Four conclusions were drawn from the results: (1) the fingerprinting and other analyses revealed significant differences in the content of each ingredient of *S. miltiorrhiza* among different processing methods. (2) The fresh processing method yielded significantly greater preservation of phenolic acids and tanshinones than the traditional processing method. (3) Among the different drying conditions during fresh processing, the shade-drying method (moisture content: 21.02–26.38%) was most beneficial for retaining the active ingredients in the roots and rhizomes of *S. miltiorrhiza*. (4) For preserving total tanshinone ingredients in particular, due to the constant temperature and short drying time, the oven-drying method (moisture content: 18.01–49.79%) might be the most suitable drying method. For preserving the contents of rosmarinic acid and salvianolic acid B, the sun-drying method (moisture content: 13.88–22.91%) might be the most suitable drying method because it is not limited by site. These results provide a reference for improving the yield of active ingredients in the industrial production of *S. miltiorrhiza.* Furthermore, we hope that the current findings lead to new ideas and insights for the processing of other TCMs.

## Figures and Tables

**Figure 1 molecules-27-04860-f001:**
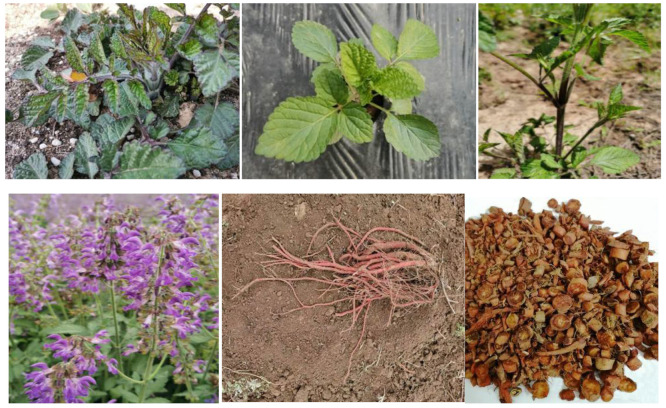
The plant and main medicinal material of *S. miltiorrhiza* used in the pharmaceutical industry. From left to right: whole plants, leaves, stems, flowers, roots and pieces.

**Figure 2 molecules-27-04860-f002:**
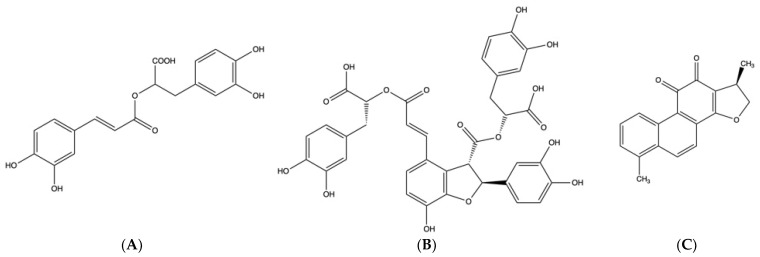
The structures of six active ingredients: (**A**)—rosmarinic acid, (**B**)—salvianolic acid B, (**C**)—dihydrotanshinone I, (**D**)—cryptotanshinone, (**E**)—tanshinone I, (**F**)—tanshinone IIA.

**Figure 3 molecules-27-04860-f003:**
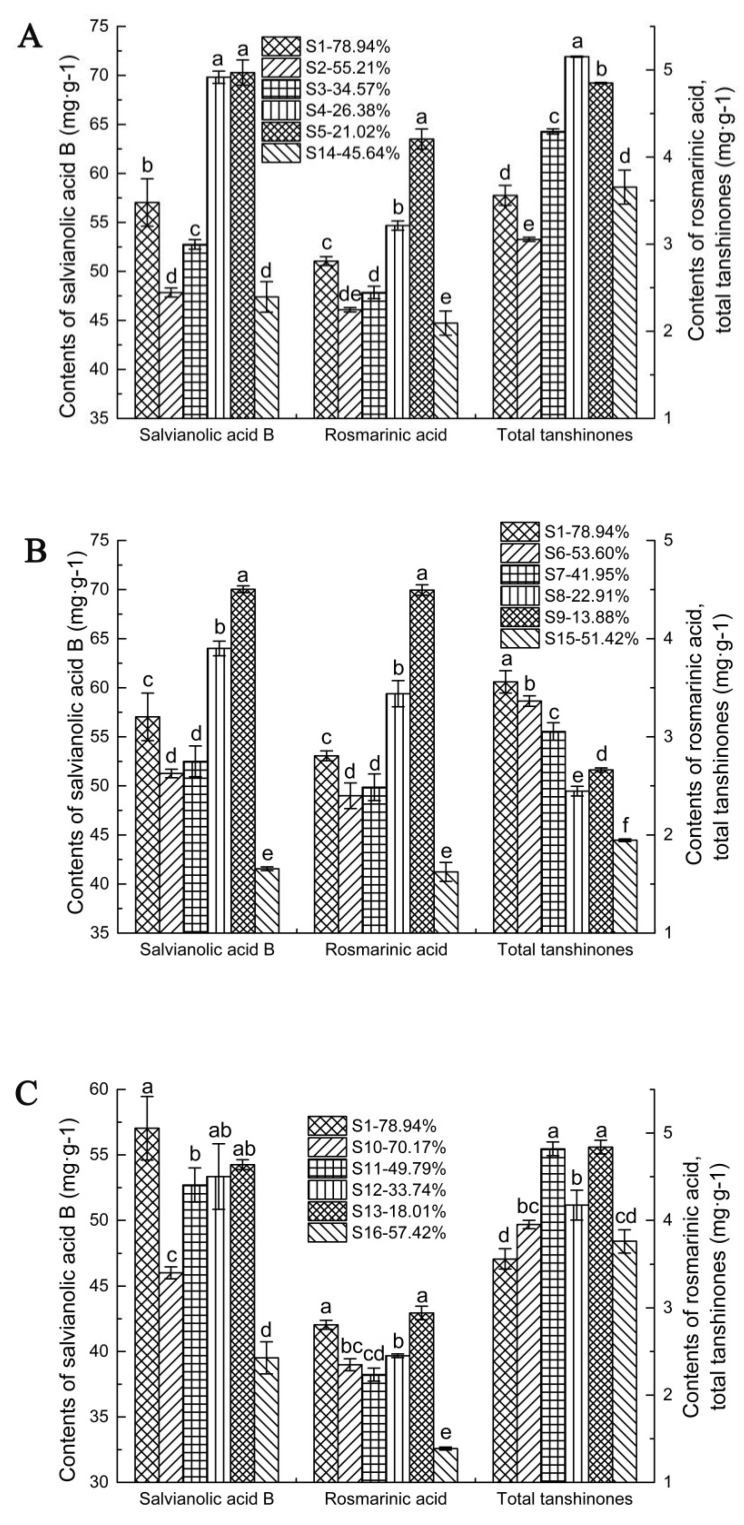
Changes in *S. miltiorrhiza* ingredient contents under different processing methods. (**A**) Shade drying; (**B**) sun drying; (**C**) oven drying. Note: *p* < 0.05.

**Figure 4 molecules-27-04860-f004:**
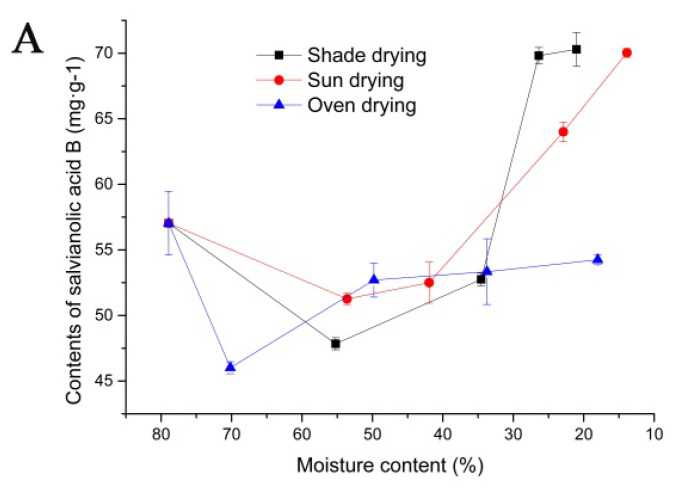
Changes in *S. miltiorrhiza* ingredient contents in different drying conditions during fresh processing. (**A**) Salvianolic acid B; (**B**) rosmarinic acid; (**C**) total tanshinones.

**Figure 5 molecules-27-04860-f005:**
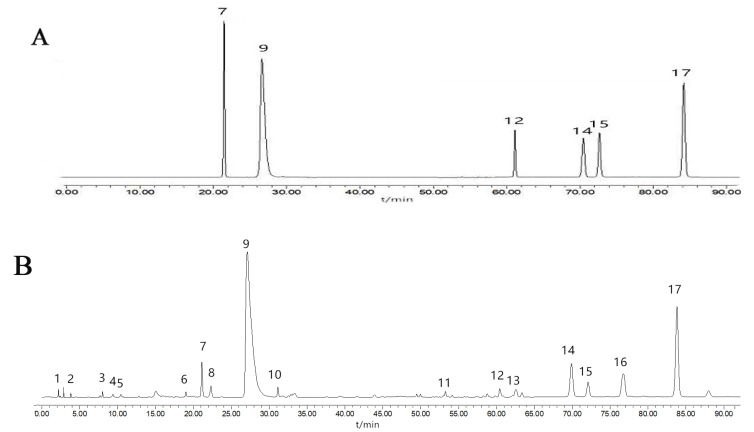
Typical chromatograms of standard mixed solution (**A**) and samples (**B**). 7-rosmarinic acid; 9-salvianolic acid B; 12-dihydrotanshinone I; 14-cryptotanshinone; 15-tanshinone I; 17-tanshinone IIA.

**Figure 6 molecules-27-04860-f006:**
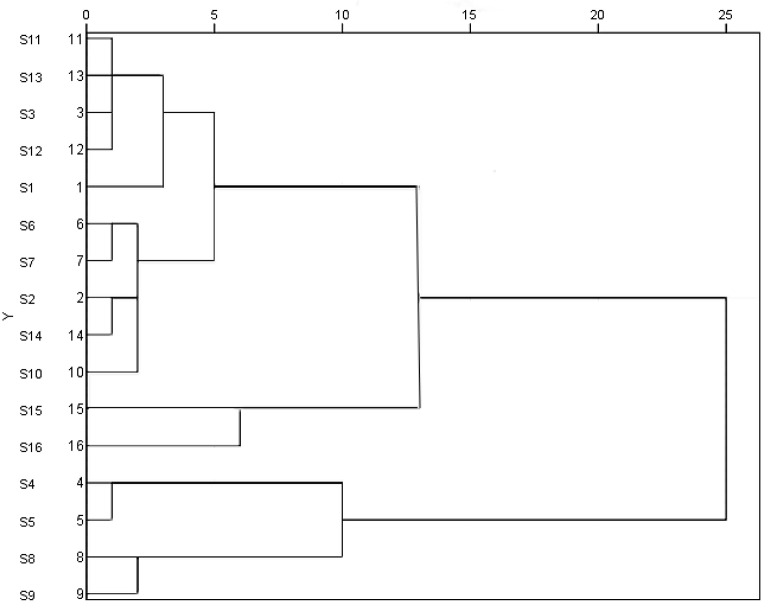
The cluster dendrogram of all active ingredients of 16 batches of different processing methods of *S. miltiorrhiza*.

**Figure 7 molecules-27-04860-f007:**
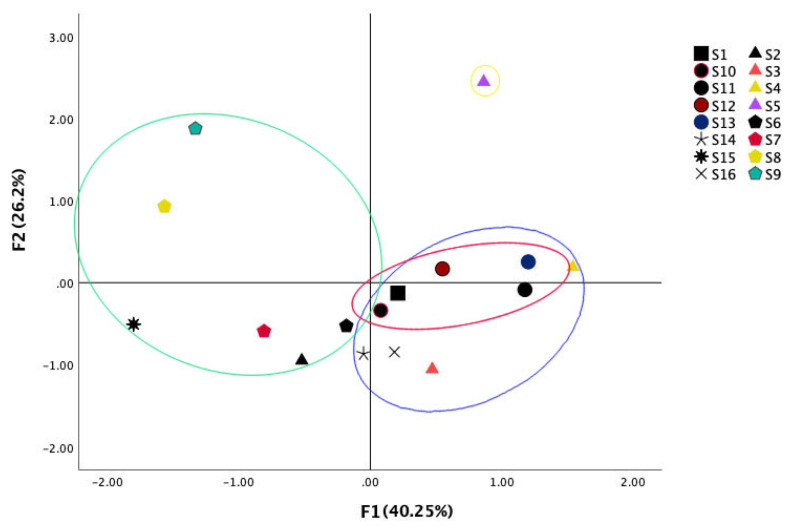
The 2D score plot of all active ingredients of 16 batches of different processing methods of *S. miltiorrhiza* with the first two three principal components.

**Figure 8 molecules-27-04860-f008:**
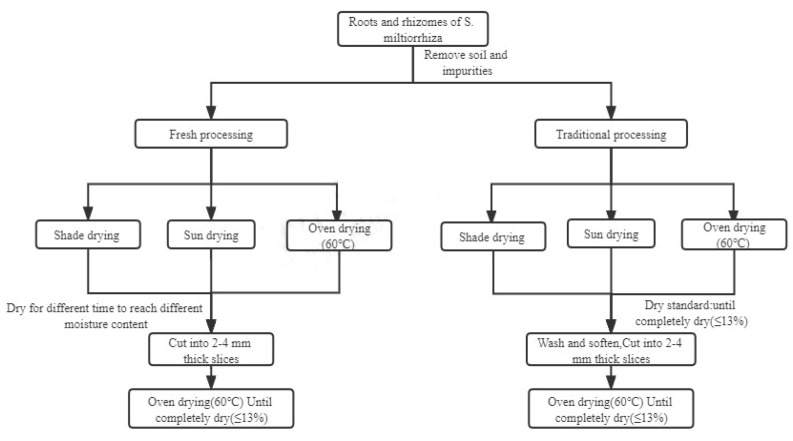
The process flow of *S. miltiorrhiza* roots and rhizome.

**Table 1 molecules-27-04860-t001:** Contents of six active ingredients in S1–S16.

Sample Code	Rosmarinic Acid (mg·g^−1^)	Salvianolic Acid B (mg·g^−1^)	Total Tanshinones (mg·g^−1^)
S1	2.806 ± 0.051 *^#&^	57.033 ± 1.706 *^#&^	3.560 ± 0.098 ^#&^
S2	2.247 ± 0.025 *	47.840 ± 0.465	3.056 ± 0.023 *
S3	2.446 ± 0.072 *	52.759 ± 0.486 *	4.294 ± 0.030 *
S4	3.214 ± 0.053 *	69.814 ± 0.620 *	5.152 ± 0.006 *
S5	4.207 ± 0.116 *	70.288 ± 1.288 *	4.850 ± 0.007 *
S6	2.399 ± 0.121 ^#^	51.263 ± 0.433 ^#^	3.364 ± 0.053 ^#^
S7	2.486 ± 0.124 ^#^	52.499 ± 1.566 ^#^	3.054 ± 0.090 ^#^
S8	3.439 ± 0.098 ^#^	64.003 ± 0.749 ^#^	2.446 ± 0.050 ^#^
S9	4.495 ± 0.054 ^#^	70.027 ± 0.346 ^#^	2.662 ± 0.020 ^#^
S10	2.348 ± 0.068 ^&^	46.016 ± 0.457 ^&^	3.952 ± 0.049
S11	2.235 ± 0.064 ^&^	52.701 ± 1.305 ^&^	4.819 ± 0.079 ^&^
S12	2.451 ± 0.021 ^&^	53.341 ± 1.604 ^&^	4.174 ± 0.121 ^&^
S13	2.942 ± 0.078 ^&^	54.250 ± 0.379 ^&^	4.839 ± 0.079 ^&^
S14	2.094 ± 0.104	47.397 ± 0.683	3.655 ± 0.181
S15	1.623 ± 0.081	41.554 ± 0.191	1.946 ± 0.014
S16	1.388 ± 0.017	39.515 ± 1.228	3.760 ± 0.102

* *p* < 0.05 vs. S14, ^#^
*p* < 0.05 vs. S15, ^&^
*p* < 0.05 vs. S16.

**Table 2 molecules-27-04860-t002:** Different processing treatment methods.

Sample Code	Raw Weight (g)	Raw Size (cm)	Raw Thickness (mm)	Drying Conditions	Moisture Content before Cutting (%)	Moisture Content after Drying (%)
S1	49.67 ± 1.13	13.50 ± 0.03	9.18 ± 0.06	No drying	78.94	6.11
S2	49.29 ± 1.47	13.59 ± 0.62	8.11 ± 0.13	Shade drying 72 h	55.21	6.23
S3	52.44 ± 0.94	14.76 ± 0.40	7.48 ± 0.10	Shade drying 120 h	34.57	4.31
S4	52.11 ± 1.51	15.17 ± 0.29	8.97 ± 0.15	Shade drying 144 h	26.38	4.20
S5	56.57 ± 0.56	15.39 ± 0.14	9.80 ± 0.02	Shade drying 192 h	21.02	4.40
S6	55.88 ± 1.03	16.85 ± 0.30	8.08 ± 0.02	Sun drying 48 h	53.60	4.19
S7	44.68 ± 1.35	14.52 ± 0.08	8.09 ± 0.06	Sun drying 72 h	41.95	7.58
S8	57.50 ± 0.72	17.84 ± 0.04	8.67 ± 0.07	Sun drying 120 h	22.91	3.92
S9	56.02 ± 1.34	14.57 ± 0.10	9.21 ± 0.11	Sun drying 144 h	13.88	4.02
S10	44.12 ± 1.27	14.58 ± 0.04	8.85 ± 0.04	Oven drying (60 °C) 4 h	70.17	5.52
S11	46.38 ± 1.24	17.46 ± 0.04	8.45 ± 0.03	Oven drying (60 °C) 8 h	49.79	5.79
S12	55.07 ± 1.58	18.57 ± 0.03	8.64 ± 0.02	Oven drying (60 °C) 12 h	33.74	5.11
S13	56.52 ± 1.10	17.17 ± 0.08	7.81 ± 0.32	Oven drying (60 °C) 16 h	18.01	4.35
S14	58.61 ± 1.92	18.19 ± 0.05	8.49 ± 0.05	Shade drying 720 h	45.64	3.60
S15	44.40 ± 1.59	17.28 ± 0.03	9.52 ± 0.16	Sun drying 480 h	51.42	3.44
S16	44.88 ± 1.47	18.43 ± 0.08	8.46 ± 0.24	Oven drying (60 °C) 48 h	57.42	5.52

## Data Availability

Not applicable.

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
