# Peer review of "Effects of Different Processing Methods Based on Different Drying Conditions on the Active Ingredients of Salvia miltiorrhiza Bunge"

_molecules, 2022, doi:10.3390/molecules27154860_

Round 1

Reviewer 1 Report

Effects of different processing methods based on different drying conditions on the active ingredients of Salvia miltiorrhiza were performed, following content of rosmarinic acid, salvianolic acid, and total tanshiones using standards and HPLC technique. One of the major concerns is given in chapter 4.3. where the experiment states that the dried samples were sieved and a total of 0.5 g of each sample was used for analysis. It is not clear how was the amount of ingredients estimated (per gram of what? dried, semi-dried?), as the used plant material contained a different percentage of moisture (see Table 2). For example, S1 represents fresh plant material without drying conditions (having 78% of moisture) and contains a similar amount of ingredients compared to the dried samples?! So, are the amounts in plant material with different drying conditions comparable per gram of the plant material used?

Some other remarks are the following:

 S. miltiorrhiza should be italic throughout the text

Figure 2 - please uniform the size of the structures

Table 1 - Please add the drying conditions in the table per sample code

Table 3 - µg mL-1

Author Response

Dear reviewers:

     We have carefully considered the suggestion of Reviewer and make some changes. We have tried our best to improve and made some changes in the manuscript.

Responds to the reviewers' comments:

Reviewer #1 

Point 1: Effects of different processing methods based on different drying conditions on the active ingredients of Salvia miltiorrhiza were performed, following content of rosmarinic acid, salvianolic acid, and total tanshiones using standards and HPLC technique. One of the major concerns is given in chapter 4.3. where the experiment states that the dried samples were sieved and a total of 0.5 g of each sample was used for analysis. It is not clear how was the amount of ingredients estimated (per gram of what? dried, semi-dried?), as the used plant material contained a different percentage of moisture (see Table 2). For example, S1 represents fresh plant material without drying conditions (having 78% of moisture) and contains a similar amount of ingredients compared to the dried samples?! So, are the amounts in plant material with different drying conditions comparable per gram of the plant material used?

Response 1: We appreciate it very much for this good suggestion. Before the contents determination, we have dried 16 batches of S. miltiorrhiza decoction pieces until the moisture contents is less than or equal to 13.0% (according to the standard of Chinese Pharmacopoeia 2020 edition). This operation has been described in chapter 4.2 of the original manuscript. And we have added the water contents (3.44-7.58%) of 16 batches of samples before contents determination in Table 2. We determine water contents by oven drying method (2020 Chinese Pharmacopoeia Part IV: 0832 moisture determination method-oven drying method). Due to the existence of experimental deviations, it is impossible to make the moisture contents of each batch of samples completely consistent after completely drying. Our study purpose is to compare the effect of different processing methods (including different drying methods and cutting S. miltiorrhiza with different moisture content) on S. miltiorrhiza. However, for S. miltiorrhiza decoction pieces before contents determination, as mentioned above, they have been completely dried. Therefore, the active ingredients contents of S. miltiorrhiza of different processing is completely reasonable. In addition, as shown in Table 1, mg· g-1 was used as the unit of active ingredients.

Point 2: S. miltiorrhiza should be italic throughout the text

Response 2:We appreciate it very much for this good suggestion, and it has been rectified throughout the text.

Point 3: Figure 2 - please uniform the size of the structures.

Response 3:We appreciate it very much for this good suggestion, and the size of the structures have been unified in figure 2.

Point 4: Table 1 - Please add the drying conditions in the table per sample code.

Response 4:We agree with you that add the drying conditions in the table per sample code. However, this may be misleading to other readers, as the different drying conditions are only part of the processing methods, and S1-S16 refers to the sample number of the whole process. This is illustrated in Table 2 and 4.2.

Point 5: Table 3 - µg mL-1

Response 5: We are very sorry for our incorrect writing and it is rectified at table B1.

Reviewer 2 Report

Authors compared two different processing method  on the content of active components in S. miltiorrhiza.  The study has some value, however,  manuscript is chaotic and needs correction (also English correction).

Main comment:

Were the results  expressed on fresh or dried weight of plant material? What was the final water content in samples taken for HPLC analysis? Without this information the scientific value of the manuscript is hardly to assessed. The differences in content of water in dried material has significant impact on the results of quantification.

Moreover:

The aim of the study in Introduction should be highlighted. Now it is just description of the study.  Conclusion here are unnecessary (line 103, 105, 109-11).

Lack of method for water content assessment.

Numbering of Figures and Tables are incorrect: e.g. line 147: should be Fig.3, Line 341, 343: should be:  table 2

“As shown in Table A1 and Table A2, the relative  tR R.S.D. of each common peak ranged from 0.036-1.640%” -  no retention times are shown in Table 1 and Table 2  

“and the relative PA R.S.D. ranged from 7.772-64.680%” – RSD 64%?

Organization of the manuscript needs improvement. Tables and Figures should be placed near the first mention in the text.

4.5. Validation – this section is unnecessary in main body and should be moved to Supplementary material.  Lack of limit of detection and quantification

Figure 4 is not informative and is unnecessary (should be moved to Supplementary)

Minor errors:

- lack of italic for the name of the plant in a lot of places

-incorrect format: „mg·g-1”

- lack  of spaces in a lot of places

Author Response

Dear reviewers:

    We have carefully considered the suggestion of Reviewer and make some changes.We have tried our best to improve and made some changes in the manuscript.

Responds to the reviewers' comments:

Main comment:

Point 1: Were the results  expressed on fresh or dried weight of plant material? What was the final water content in samples taken for HPLC analysis? Without this information the scientific value of the manuscript is hardly to assessed. The differences in content of water in dried material has significant impact on the results of quantification.

Response 1: We appreciate it very much for this good suggestion, the results were expressed on dried weight of plant material. The water contents of completely dried S. miltiorrhiza are shown in Table 2. This operation has been described in chapter 4.2 and Figure 8 of the original the revised manuscript.

Moreover:

Point 2: The aim of the study in Introduction should be highlighted. Now it is just description of the study.  Conclusion here are unnecessary (line 103, 105, 109-11).

Response 2: Thank you very much for your advice. The aim of our study is very clear - Evaluating the differences between fresh processing and traditional processing, and demonstrated the feasibility of fresh processing. Meanwhile, the differences in the contents of active ingredients among different drying conditions under fresh processing were studied. Line 103 and 105 are the aim of the study, and we have deleted the conclusion in the introduction.

Point 3: Lack of method for water content assessment.

Response 3: We are very sorry for our incorrect and the method for water content assessment have been added and identified its source in 4.5 of Revised manuscript. (2020 Chinese Pharmacopoeia Part IV: 0832 moisture determination method-oven drying method).

Point 4: Numbering of Figures and Tables are incorrect: e.g. line 147: should be Fig.3, Line 341, 343: should be:  table 2

Response 4: We are very sorry for our incorrect writing. We have made modifications in the revised manuscript.

Point 5: “As shown in Table A1 and Table A2, the relative  tR R.S.D. of each common peak ranged from 0.036-1.640%” -  no retention times are shown in Table 1 and Table 2  

Response 5: Thank you for your suggestion. The 17 common peaks in Table A 1 and A 2 contain the peaks of the active ingredients in Table 1, and the R.S.D. of each active ingredient has been obtained in 4.5.2, as shown in Table B2 and Table B3. This proved that the method we used and the active ingredients to be measured were applicable in instrumental HPLC. Based on this, we determined the contents of the active ingredients in each batch of samples and presented it in Table 1.

Point 6: “and the relative PA R.S.D. ranged from 7.772-64.680%” – RSD 64%?

Response 6: In S1-S16 samples, the relative PA R.S.D. of the same peak was much higher than 5%, which indicated that the contents of all components in different batches of S. miltiorrhiza was greatly different, proving the significance of our study.

Point 7: Organization of the manuscript needs improvement. Tables and Figures should be placed near the first mention in the text.

Response 7: We are very sorry for our mistake and thank the reviewer for reminding us. We have corrected this problem.

Point 8: 4.5. Validation – this section is unnecessary in main body and should be moved to Supplementary material.  Lack of limit of detection and quantification

Response 8: This is a good suggestion and I have put chapter 4.5 in the supplementary materials section. The methodological verification in HPLC was carried out in full accordance with the verification indexes stipulated by The Chinese Pharmacopoeia. The verification of the limits of quantitation and detection mentioned is required only in the determination of impurities.

Point 9: Figure 4 is not informative and is unnecessary (should be moved to Supplementary)

Response 9: Figure 4 provides the fingerprint information of 16 batches of S. miltiorrhiza samples processed in different ways, which I have put into the supplementary materials.

Point 10: Minor errors:

- lack of italic for the name of the plant in a lot of places

-incorrect format: „mg·g-1”

- lack  of spaces in a lot of places

Response 10: Thank you for your advice. We have modified the above minor errors.

Round 2

Reviewer 1 Report

The authors have answered all the questions and made appropriate corrections and thus can be accepted for publication.

Author Response

Dear reviewer:

     Thank you for the reviewers’ comments concerning our manuscript entitled “Effects of different processing methods based on different drying conditions on the active ingredients of Salvia miltiorrhiza Bunge” (ID: molecules-1822341). The english language and style have been checked and revised.

Reviewer 2 Report

1) Response: The aim of our study is very clear (…)

I understand the aim of Your study but in scientific paper usually the form: „the aim of our study was…”  is used or  research hypothesis is described. Moreover: „The shade drying method was found to be most conducive to the retention of active ingredients.” – this is a conclusion and it should be removed from Introduction (the same comment for „which revealed significant differences among the methods.”

2) Response: The verification of the limits of quantitation and detection mentioned is required only in the determination of impurities.

Maybe Chinese Pharmacopoeia requires to assess LOD and LOQ only for impurities but European standards (see ICH guideline) recommends to determine LOD and LOQ for all validation procedures. Thus, LOD and LOQ should be added if Authors stated that method was validated.  However, because this section was removed from main body of the manuscript  it is not mandatory now. This is a comment for future projects.

3) relative PA R.S.D/ relative tR R.S.D – the word „relative” is unnecessary. RSD means „relative standard deviation”

4) Supplementary material in Molecules is usually provided as separate file (not included in main body).

5) There are still a lot of typo errors:

- lack of italic for the name of the plant in a lot of places (e.g. line 85, 134)

-incorrect format: „mg·g-1” – see Tables and Figures

- lack of spaces in a lot of places (e.g. line 67,85, 49…..)

Author Response

Dear reviewers:

       Thank you for the reviewers’ comments concerning our manuscript entitled “Effects of different processing methods based on different drying conditions on the active ingredients of Salvia miltiorrhiza Bunge” (ID: molecules-1822341). Those comments are all valuable and very helpful for revising and improving our paper, as well as the important guiding significance to our researches. We have studied comments carefully and have made correction which we hope meet with approval. Revised portion are marked in red in the paper. The main corrections in the paper and the responds to the reviewer’s comments are as flowing:

Point 1: I understand the aim of Your study but in scientific paper usually the form: „the aim of our study was…”  is used or  research hypothesis is described. Moreover: „The shade drying method was found to be most conducive to the retention of active ingredients.” – this is a conclusion and it should be removed from Introduction (the same comment for „which revealed significant differences among the methods.”

Response 1: We appreciate it very much for this good suggestion, We have modified the format of the study to "the aim of our study was..." according to your suggestion, we also deleted "The shade drying method was found to be most conducive to the retention of active ingredients.”and“which revealed significant differences among the methods.”

Point 2: Maybe Chinese Pharmacopoeia requires to assess LOD and LOQ only for impurities but European standards (see ICH guideline) recommends to determine LOD and LOQ for all validation procedures. Thus, LOD and LOQ should be added if Authors stated that method was validated.  However, because this section was removed from main body of the manuscript  it is not mandatory now. This is a comment for future projects.

Response 2: Thank you very much for your advice. We will value the assess of LOD and LOQ for all validation procedures in the future.

Point 3:relative PA R.S.D/ relative tR R.S.D – the word „relative” is unnecessary. RSD means „relative standard deviation”

Response 3: Yes, RSD means relative standard deviation but the 'relative' in relative PA / relative tR uses the seventeenth peak of the common peak (Tanshinone IIA) as the reference peak to calculate the relative peak area and relative retention time of other peaks, not the peak area and the relative standard deviation of PA and tR.

4) Supplementary material in Molecules is usually provided as separate file (not included in main body).

Response 4: Thank you very much for your advice. We have moved the Supplementary material in Molecules to the Supplementary file.

5) There are still a lot of typo errors:

- lack of italic for the name of the plant in a lot of places (e.g. line 85, 134)

-incorrect format: „mg·g-1” – see Tables and Figures

- lack of spaces in a lot of places (e.g. line 67,85, 49…..)

Response 5: We are very sorry for our mistake and thank the reviewer for reminding us. We have corrected these problems and check other errors in article.